# Experimental Study on Flexural Bearing Characteristics of Ceramsite Concrete Beams after Creep at Different Curing Ages

Hongbing Zhu [1,2,*], Jingyi Chen [1], Yuxin Wu [1], Jiapeng Li [1,3], Zhenghao Fu [1] and Peng Liu [4]

1 School of Urban Construction, Wuhan University of Science and Technology, Wuhan 430065, China
2 Institute of High-Performance Engineering Structure, Wuhan University of Science and Technology, Wuhan 430065, China
3 The National Prefabricated Construction Industry Base, Huasen Architecture and Engineering Design Consulting Company, Shenzhen 518000, China
4 School of Civil Engineering, Central South University, 22 Shaoshan Road, Changsha 410075, China
* Correspondence: zhuhongbing@wust.edu.cn

**Abstract:** Ceramsite concrete is increasingly used for structural applications due to its lightweight, high-strength, and high-temperature resistance advantages. However, research into the postcreep properties of ceramsite concrete has yet to keep pace with other areas. Therefore, this paper aims to investigate the flexural bearing properties of ceramsite concrete beams after creep and make recommendations for future research. To study the flexural load-bearing performance of ceramsite concrete beams after creep, a comparison test was carried out on four ceramsite concrete beams subjected to 200 d creep loading and four beams subjected to static load without creep loading for four curing ages. The results showed that, at first, the deformation capacity of the ceramsite concrete beams would increase after creep and the toughness and energy absorption capacity would also strengthen. Secondly, the creep could improve the ultimate bearing capacity of the ceramsite concrete beams. It was an approximate 10% increase after 200 d creep. Then, the stiffness, toughness, and ultimate bearing capacity of the ceramsite concrete beams would increase gradually with the increase of the curing age after the creep. In addition, the ultimate bearing capacity of the ceramsite concrete beams increased gradually with the increase of the curing age. The growth rate was faster in the early stage, slower after 120 d, and slower and more stable after 228 d. Finally, creep could cause the maximum crack width and depth of the ceramsite concrete beams decreased and the number of cracks would grow with the curing age. The conclusions obtained in the study provide a theoretical basis for the design of light aggregate structures for creep.

**Keywords:** ceramsite concrete beam; crack; creep; curing ages; ultimate bearing capacity

## 1. Introduction

In recent years, as the spans of buildings and bridges have increased, the self-weight of structures is approaching the limits of what ordinary concrete can withstand. As a result, reducing the weight of such systems has attracted increasing research attention [1].

Ceramsite concrete can reduce the weight of ordinary concrete by 20% to 40% under the same load-bearing conditions [2,3]. Due to its lightweight, high strength, and high temperature resistance [4], it has been used in high-rise buildings and bridge projects and is of high research value. In practical engineering, most ceramsite concrete structures are subjected to long-term loading, causing creep in the structure. Jiang [5] found that ceramsite concrete has greater creep values than ordinary concrete, which can be up to 1.3 times greater. Under long-term loading conditions, this can lead to large creep deformations and relaxations. On the one hand, this may cause the structure to tilt and deform. On the other hand, excessive structural deformation may change the structure's mechanical properties and seriously affect its safety. Both aspects deserve attention.

For them, the former relates to the laws of concrete creep and the latter to the effects of concrete creep. Currently, many studies reflect concrete creep, and the creep values of ceramsite concrete are higher than those of ordinary concrete of the same age. The tensile creep of highly porous ceramsite concrete will be lower than that of low-absorbent ceramsite concrete, and plain ceramsite concrete [6,7]. In addition, the loading age of the structure [8–10], the curing age [11] and curing temperature [12], the concrete's collapse [13] and sand content [14], dosing of steel fibres [15,16], dosing of polymeric fibers [17], admixture [18], and the relative humidity of the working environment in which the concrete elements are placed [19,20] have all been shown to be influential factors in creep. As research continued, Zhu, Glanville [21,22] and others used experimental and theoretical hypotheses on creep to propose a creep model containing loading age and loading time parameters, respectively. Compared with the studies on the creep law of concrete, there are fewer studies involving the structure's mechanical properties after creep. Dong, Rong, and others [23–25] have suggested that long-term loading causes stress relaxation in concrete beams and reduces the stress at the tip of concrete cracks, resulting in an increase in the residual bearing capacity of the concrete beam and an improvement in crack development. However, Liu [26] obtained the opposite test results. He found that reclaimed concrete beams under long-term loading exhibited postductility and that secondary loading reduced their flexural bearing capacity. Furthermore, Saliba and Omar [27,28], among others, suggested that the development of microcracks in concrete during creep can reduce concrete residual strength. It is challenging to provide scientific references for practical engineering. Therefore, the effect of creep on the mechanical properties of ceramsite concrete structures needs to be urgently investigated. In addition, in existing studies, extending the loading age of plain concrete beams (i.e., increasing the curing age) reduces the effect of creep on the structure [8]. This gives good ideas for improving the mechanical properties of ceramsite concrete structures after creep. This is also a technical issue that deserves attention.

Analyzing the above, ceramsite concrete has good prospects for engineering applications because of its lightweight and high-strength properties. However, the effect of creep on its mechanical properties has yet to be unified, which does not provide a scientific reference for practical engineering. In addition, although extending the age of curing can reduce the effect of creep on plain concrete beams, it is still being determined whether it can improve the mechanical properties of ceramsite concrete beams after creep. Clarifying and improving the effect of creep on the mechanical properties of ceramsite concrete structures is a significant issue. Therefore, this study aims to clarify the effect of creep on the flexural properties of ceramsite concrete beams. On this basis, the influence of curing age on flexural bearing characteristics of ceramsite concrete beams after creep is found. In order to achieve these objectives, the following work was carried out in this paper. (i) Eight ceramsite concrete beams with four types of curing ages were produced. Then, the loading of 30% flexural ultimate bearing capacity was carried out on one group of them (four beams with different curing ages) for 200 days. Next, the two groups of the beam were tested under various loads by step loading, and the cracking load, failure load, and crack development of the beam were recorded. (ii) Based on the deformation law, ultimate bearing capacity, beam damage, and crack distribution, the influence of long-term creep on the flexural performance of ceramsite concrete beams was analyzed by comparing them with the experimental beams of the same age without creep loading. (iii) We drew the influence curve of ultimate bearing capacity and crack distribution curve of ceramsite concrete beams after creep with different curing ages during a static load test and fitted the data to analyze the influence of curing ages on mechanical properties of ceramsite concrete beams after creep.

## 2. Experimental Materials and Basic Mechanical Properties

### 2.1. Experimental Materials

So that related workers can repeat the experiment in the future, all materials used are listed as follows.

(1)   The coarse aggregate is 900-grade crushed stone shale granules produced by Yichang Guangda Ceramic Granules Products Co. Prewetting ceramsite before concrete mixing can effectively improve the workability of concrete. Its physical properties are shown as Table 1. The ceramsite is prewetted for 4 h, and then dried for 12 h to remove the water on the ceramsite surface (shown in Figure 1).

**Table 1.** Physical properties of shale ceramsite.

| Particle Size (mm) | Volume Density (kg/m³) | Apparent Density (kg/m³) | Compressive Strength of Concrete Cylinder (MPa) | Water Absorption in 1 h (%) |
|---|---|---|---|---|
| 5~20 | 814 | 1517 | 6.8 | 2.42 |

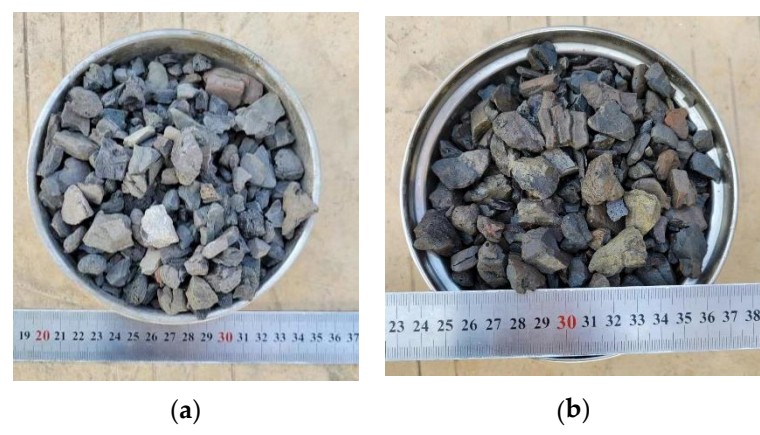

(**a**)                                                            (**b**)

**Figure 1.** Shale ceramsite. (**a**) Before wetting process. (**b**) After wetting process.

(2)   Fine aggregate was ordinary river sand, which was sieved before the test to avoid the influence of impurities in the sand. The filter sieve used for the test was a 4.75 mm side length square hole sieve. The physical properties are shown in Table 2.

**Table 2.** Physical properties of sand.

| Volume Density (kg/m³) | Apparent Density (kg/m³) | Mud Ration (%) | Fineness |
|---|---|---|---|
| 1570 | 2650 | ≤2 | 2.7 |

(3)   The cement was P.O. 42.5 ordinary silicate cement produced by Hubei Huaxin Cement Company Limited, Wuhan, China and its physical indices all met the requirements of the test specification, as shown in Table 3.

**Table 3.** Basic physical index of cement.

| Density (g/cm³) | Mineral Composition of Clinker | | | | Fineness(Sieve Allowance by 80 μm Square Hole)/% |
|---|---|---|---|---|---|
| | C3S | C2S | C3A | C3AF | |
| 3.15 | 45 | 25 | 12 | 8 | 6.5 |

(4)   The water-reducing agent is HsC polycarboxylat-type superplasticizers produced by Qingdao Hongxia. The technical specifications of the water-reducing agent are shown in Table 4.

**Table 4.** Technical index of the water-reducing agent.

| Color | PH | Relative Density | Solid Content (%) | Water-Reducing Rate (%) |
|---|---|---|---|---|
| Pale yellow | 6~8 | $1.08 \pm 0.02$ | 40 | 25–35 |

(5)   The water was Wuhan tap water.

*2.2. Mix Proportion*

The strength of the concrete prepared for this test was based on the actual bridge project and the proposed strength class was LC40 ceramsite concrete. Referring to the technical standard for applying lightweight aggregate concrete (JGJ/T 12-019) in China [29], LC40 was used as the target of the trial formulation. Based on the research of the former research group [30], the mix proportion of ceramsite concrete with the strength grade of LC40 was determined after several trial mixes, as shown in Table 5. On the basis of this mix proportion, the density of ceramsite concrete was 1942 kg·m$^{-3}$.

**Table 5.** Mixed proportion of ceramsite concrete beam specimens (unit: kg/m$^3$).

| Cement | Ceramsite | Sand | Water-Reducing Agent | Water | Sand Ratio |
|---|---|---|---|---|---|
| 510 | 554 | 730 | 5.1 | 152 | 43% |

*2.3. Basic Mechanical Properties of Concrete*

According to GB/T50081-2019 Standard for Physical and Mechanical Properties of Concrete [31], the compressive strength test used 100 mm × 100 mm × 100 mm cube test blocks. The axial compressive strength test used 100 mm × 100 mm × 300 mm prismatic test blocks. Flexural strength test use 100 mm × 100 mm × 400 mm prismatic test blocks. The test blocks are made from the mix proportions shown in Table 5. Tests were carried out after 28 d of curing to obtain their cubic compressive strength, axial compressive strength and flexural strength values. The specimens cured for 28 d were also tested for static compressive modulus of elasticity and the static compressive modulus of elasticity of ceramsite concrete was measured. The results are shown in Table 6.

**Table 6.** Basic mechanical indices of ceramsite concrete after 28-d maintenance.

| Mechanical Parameters | Strength | | | |
|---|---|---|---|---|
| | Specimen No. 1 | Specimen No. 2 | Specimen No. 3 | Test Results |
| Compressive strength/MPa | 40.04 | 38.77 | 42.47 | 40.43 |
| Flexural strength/MPa | 6.92 | 7.05 | 6.43 | 6.80 |
| Elasticity modulus/GPa | 24.45 | 24.01 | 24.18 | 24.21 |

## 3. Design of the Test

This research aimed to study the change in the flexural capacity of ceramsite concrete under creep. Based on this goal, 12 specimens were divided into three groups.

The ceramsite concrete beams used in the test were designed depending on the principle of "strong shear and weak bending". Their cross-section size were 100 mm × 150 mm and span length *L* were 1500 mm. In addition, their net span L0 were 1400 mm. The longitudinal reinforcement of the beam was 24. The erection reinforcement was 48, type HRB400. the hoop reinforcement was φ6, type HPB300. Furthermore, the thickness of the concrete protective layer was 20 mm, as shown in Figure 2.

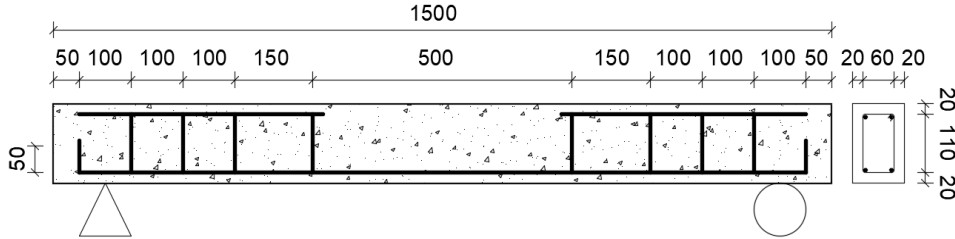

**Figure 2.** Structure of ceramsite concrete beam (unit: mm).

A total of 12 test beams were poured. To confirm the creep loading value of the test beam at each curing age, four beams were tested for the ultimate bending-bearing capacity under static load at 28 d, 60 d, 90 d, and 120 d of natural curing, respectively. As a reference group for the creep test beams, four beams were subjected to static load damage tests after 228 d, 260 d, 290 d, and 320 d of natural curing. The last four beams were subjected to 200 d of creep loading after 28 d, 60 d, 90 d, and 120 d of natural curing. Moreover, test setup for sustained load for reference Li [16], the four-point loading method was used for surcharge loads (shown as Figures 3 and 4).

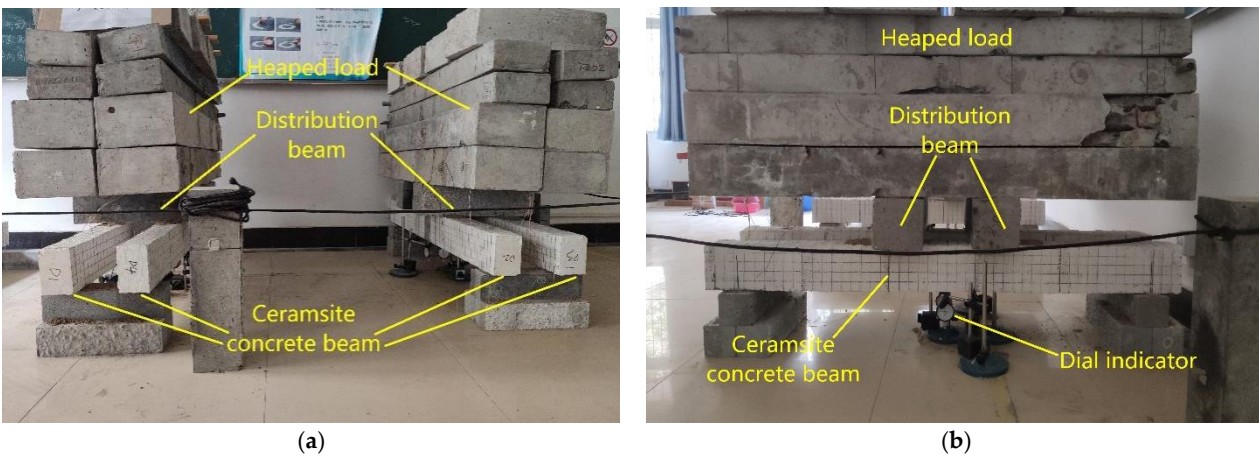

| (a) | (b) |

**Figure 3.** Snapshots of loading. (**a**) Front view. (**b**) Side view.

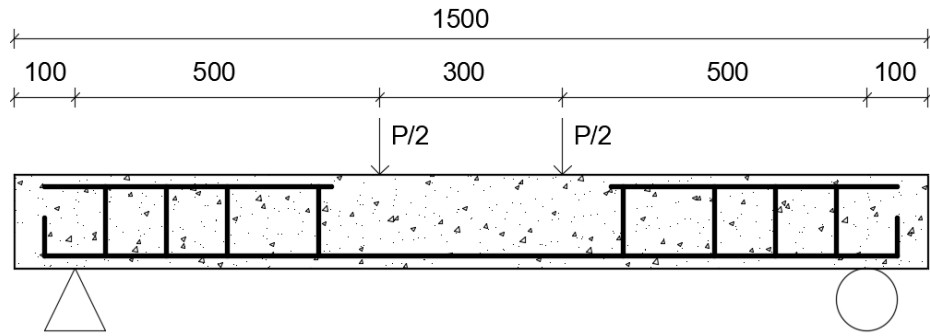

**Figure 4.** Diagram of loading (unit: mm).

This experiment takes into account laboratory specifics and safety issues, placing two test beams side by side. As the loading forces required for each beam are different, the following approach was taken for the experiment in order to combine the data from Table 7 with Figure 3. (i) First, we weigh and prestack each weight. We place a jack at the loading point of the distribution beam and then place the weight evenly on the distribution beam so that the force acting on the jacks are the same as the loading force shown in Table 7. (ii) Secondly, we mark in detail the distribution beam and the location of the jacks reaction points. (iii) Thirdly, the jacks are removed and the support, distribution beams and weight

are placed in the marked positions. Referring to existing studies [14,32,33], the loading stress is taken to be 30% of the ultimate load of the specimen beams. the spanwise deflection of the specimen beams are measured with dial indicators during the test and the spanwise deflection values of the specimen beams are recorded periodically.

**Table 7.** Loading parameters of test beam.

| Group No. | Curing Age/d | Loading Duration/d | Purpose | Ultimate Bearing Capacity/kN | Loading Force/kN | Ultimate Bearing Capacity after Creep/kN |
|---|---|---|---|---|---|---|
| L0D1 | 28 | | | 48.0 | – | – |
| L0D2 | 60 | 0 | Used to confirm creep loading force. | 50.0 | – | – |
| L0D3 | 90 | | | 52.5 | – | – |
| L0D4 | 120 | | | 53.5 | – | – |
| L1D1 | 228 | | | 55.9 | – | – |
| L1D2 | 260 | 0 | The reference group of creep test beams. | 56.4 | – | – |
| L1D3 | 290 | | | 56.8 | – | – |
| L1D4 | 320 | | | 57.2 | – | – |
| L2D1 | 228 | | Control group | – | 14.40 | 61.9 |
| L2D2 | 260 | 200 | to study the | – | 15.00 | 61.8 |
| L2D3 | 290 | | influence of | – | 15.75 | 62.8 |
| L2D4 | 320 | | curing age. | – | 16.05 | 63.1 |

After the creep test, the flexural bearing capacity of ceramsite concrete beams was tested. The test was carried out by using four-point loading and the loading schematic is shown in Figure 5. Strain gauges were applied at the beam's midspan before the loading started. The strain values were measured during loading to verify the flat section assumption. First, we check whether there was a defect in the sensitive grid was checked and the multimeter was used to check whether there was a short-circuit phenomenon. Then, the surface of the concrete where the strain gauges were to be applied was polished with abrasive paper along the 45° direction of the strain gauges, cleaned with alcohol and scribed and positioned. Finally, strain gauges were applied. In addition, a layer of AB glue was evenly applied to the strain gauges to protect them. Before the loading test, they were allowed to dry naturally for one day.

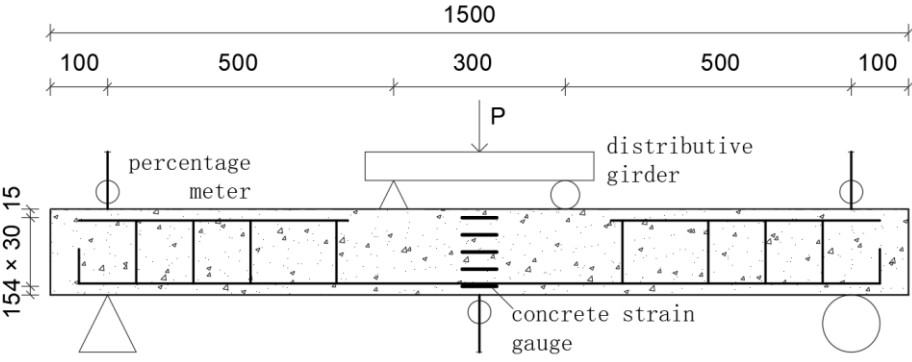

**Figure 5.** Diagram of static load test (unit: mm).

After the test preparation, a percentage meter was installed at the support and midspan of the ceramsite concrete beam to measure the midspan deflection. According to GB/T 50152-2012 Standard for Test Methods of Concrete Structures [34], preloading was carried out before formal loading. Moreover, the location of the specimens and instruments were checked to see if they were normal. The preload was carried out three times with continuous specimen position adjustment and instrumentation during the test. After passing the inspection, the official loading was performed.

The static load loading equipment of the test beam was a 200-t electrohydraulic servo pressure tester. The concrete strain value was collected by a static resistance strain gauge and the deflection was measured by a percentage meter. In addition, step-by-step loading was adopted. Each step's loading amount was 5 kN and the loading speed was 0.5 kN/s. When loading to 20 kN, the loading amount was adjusted to 2 kN per stage and the loading speed was set in 0.2 kN/s. When loading to 40 kN, the loading amount was adjusted to 1 kN per stage. The loading speed was controlled at 0.1 kN/s. Although the load value reached the ultimate bearing capacity of the ceramsite concrete beam, the pressure tester unloaded automatically. During the test, the loading time of each load level was controlled at 1–3 min. The midspan deflection and strain values imported into the computer were recorded and the crack development was observed. The field loading diagram is shown in Figure 6.

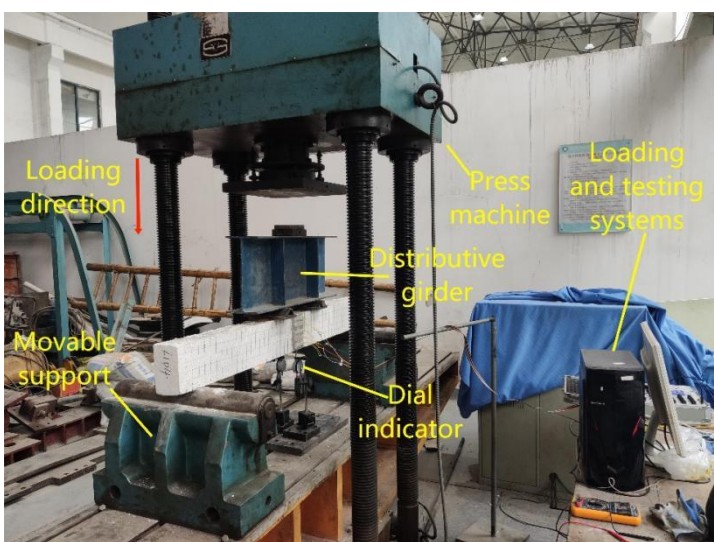

**Figure 6.** Snapshot of static load test.

Specimen beams L1D1, L1D2, L1D3 and L1D4 were subjected to direct static load damage tests when they were curing to their respective initial loading ages. Fore specimen beams L2D1, L2D2, L2D3, and L2D4, were first subjected to a 200-d period of creep loading at a value of 0.3 Pu when they were curing to their respective initial loading ages. Then, they were subjected to static load damage tests after the creep loading period.

## 4. Experimental Results and Analysis

### 4.1. Effect of Ceramsite Concrete Beam Performance after Creep

Figure 7 shows a typical damage pattern of ceramsite concrete beam. The cracks response of the beams were captured by the naked eye during the tests. It can be seen that all beams exhibit a similar cracking pattern whether they experience creep or not. That is, the first cracks in all volumes were produced in the bending region at the beginning of the loading. As the load is further increased, the cracks extend to the compressive side and additional cracks are produced in the shear zone. Compared with L1-type beams, L2-type beams displayed more cracks and less crack spacing. Moreover, the maximum crack width generated was also smaller during failure. In terms of the failure process, with the similar to ordinary concrete beams, both types of beams also experienced three stages. They were elastic working stage, elastic-plastic working stage and specimen failure. Figure 8 shows the load midspan deflection curves of beams L1D1 and L2D1 at static load loading. As can be seen from Figure 8, the direction and the changing trend of the load-mid span deflection curves of the ceramsite concrete beams were approximately the same, regardless of whether they had experienced creep loading. The load-deflection curves of the ceramsite concrete beams were essentially linear before the damage, with obvious yield points and roughly similar slopes of the curves. It exhibited that the preapplication of long-term loads did not

result in an improved ability of the ceramsite concrete beams to limit their deformation. However, compared with L1D1 and L2D1 showed a higher strength of 11% but 40% lower stiffness. The midspan deflections in the yielding and failure states were raised by 85% and 82%, respectively. It was shown that the preapplication of long-term loads led to enhance flexural strength and reduced static stiffness of ceramsite beams. Furthermore, ceramsite concrete beams that had experienced creep had more significant midspan deflection than beams that had not experienced creep when they reached yielding or failure. as referred to in reference [35].

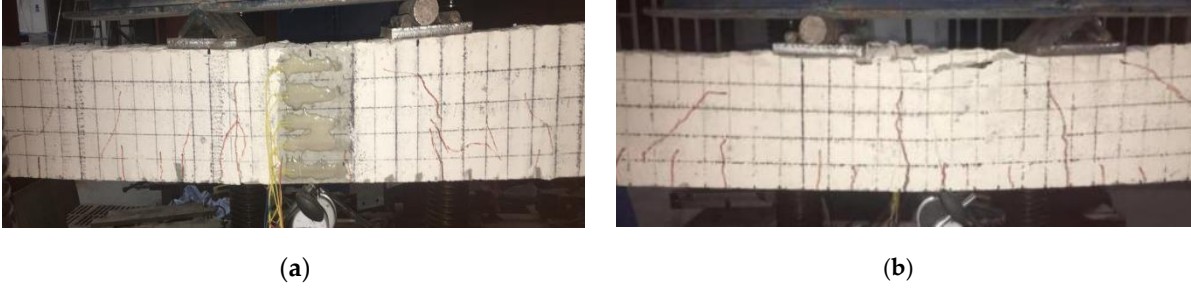

(**a**)                (**b**)

**Figure 7.** Ceramsite concrete beam damage pattern. (**a**) L1 beam damage pattern. (**b**) L2 beam damage pattern.

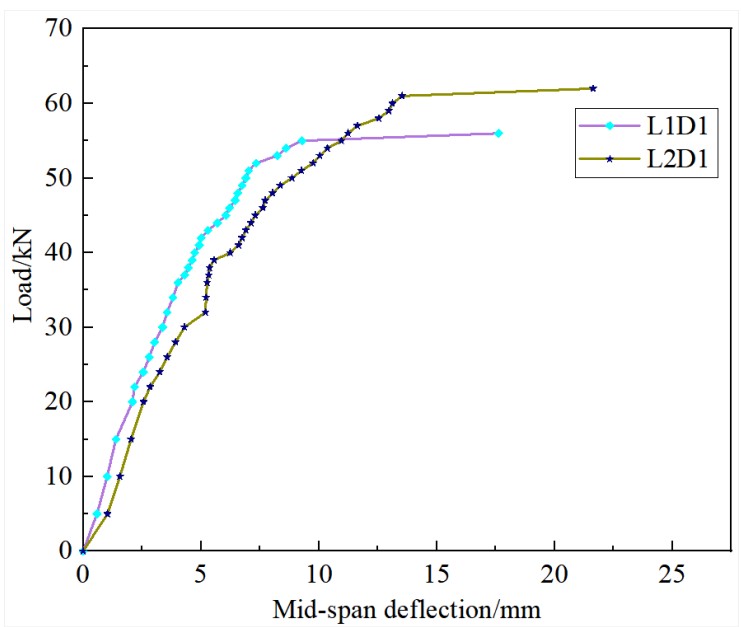

**Figure 8.** Load-deflection curve of L1D1, L2D1.

Figure 9 suggests the histogram of the ultimate bearing capacity of each beam. As shown in Figure 9, the ultimate bearing capacity of L2 beams enhanced by about 10% compared with L1 beams at the same curing age. It illustrated that long-term creep had an enhancing effect on the ultimate bearing capacity of ceramsite concrete beams. There were two plausible explanations for this experimental phenomenon [28,36]. On the one hand, it was caused by strengthening the concrete pressure zone during the long-term loading process. On the other hand, the stress relaxation reduced the stress at the tip of the concrete crack, resulting in the ultimate bearing capacity of the concrete beam being increased.

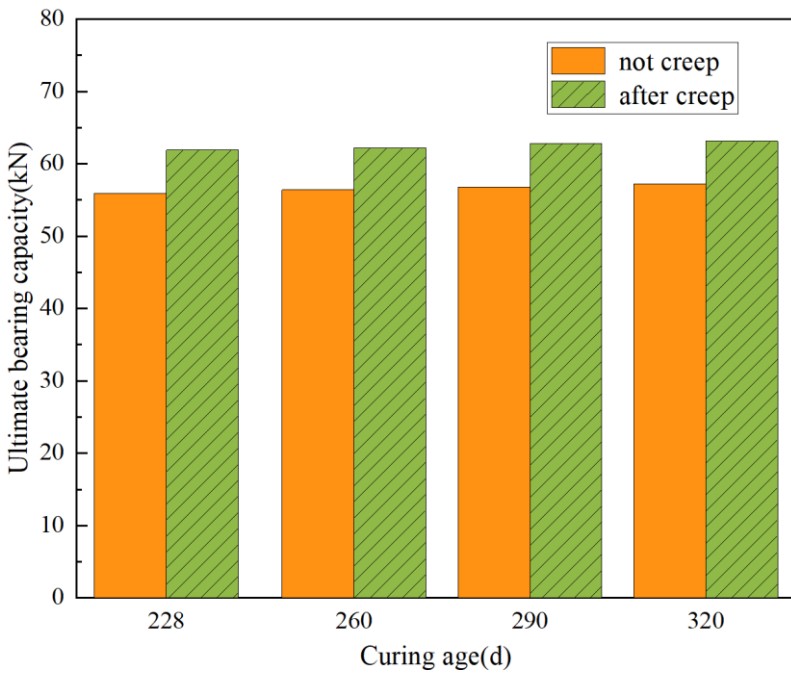

**Figure 9.** Ultimate bearing capacity of ceramsite concrete beams.

Figure 10 displays the distribution of cracks in ceramsite concrete beams. In the process of static loading, small vertical cracks appeared in the beam when it reached the cracking load. With continuously growing load values, the crack width, depth, and the number of cracks in the beams gradually increased. When beam reached the ultimate load, maximum cracking completely destroyed it.

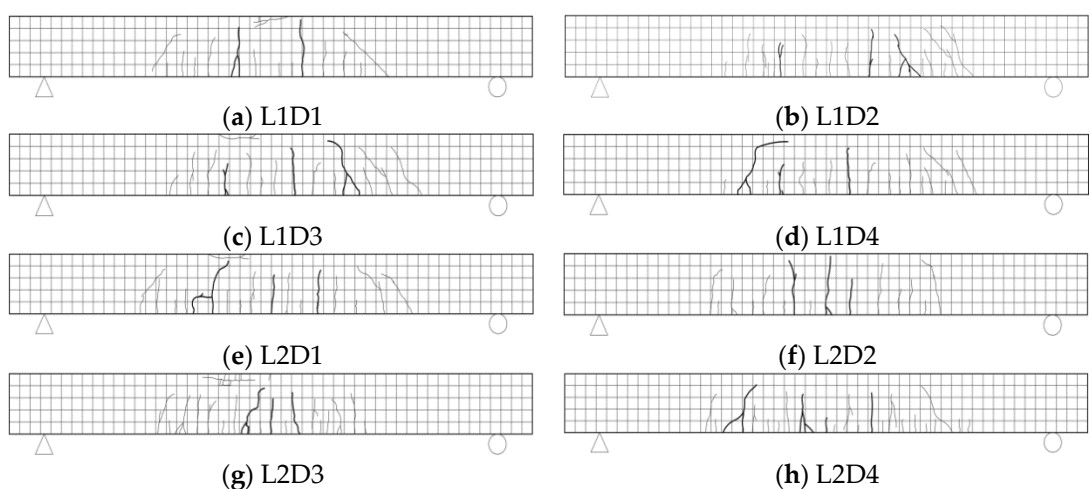

**Figure 10.** Distribution of cracks in ceramsite concrete beams.

Comparing the crack development of each test beam in Figure 10, it can be seen that the long-term creep effect caused the maximum crack width, crack depth, and average crack spacing to decrease when the ceramsite concrete beam (group L2) was damaged. Moreover, the number of cracks would increase. Because long-term loading causes stress relaxation in the concrete in the tension zone, stresses at the crack tip were reduced. In addition, cracks in the tension zone were difficult to develop when larger loads are applied. These could lead to that the maximum crack width and crack depth of ceramsite concrete beams were not significant when subjected to bending damage. When the ceramsite concrete beam reached the cracking load, although the crack width and the crack depth could not

be further developed, the number of cracks would continue to increase. In addition, the average crack spacing remains the same [37].

Figure 11 exhibits the load-span deflection curves for the L1 and L2 series test beams when applying static load damage. As shown in Figure 11, the midspan deflection of the test beams in both the yielded and damaged states at the same age after casting with static load damage had a significant increase after creep.

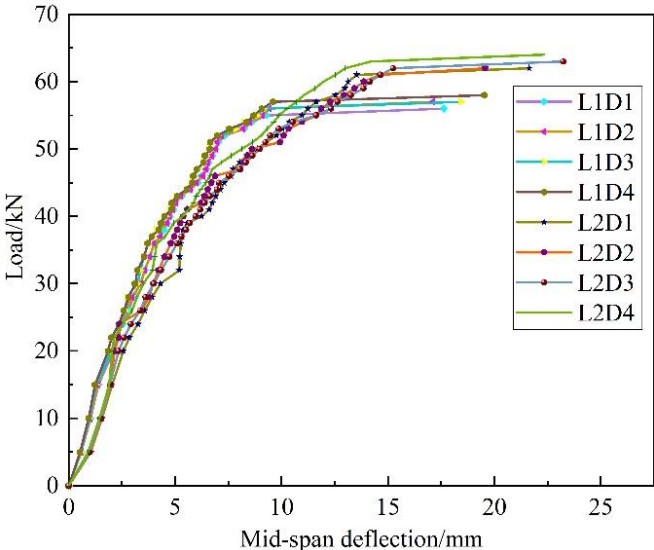

**Figure 11.** Load midspan deflection curve of ceramsite concrete beam in static load test.

It can also be seen from Figure 11 that the flexural stiffness of a beam undergoing creep enhanced as the age of caring increased. The deflection in the span corresponding to the transformation from elastic to plastic deformation became progressively larger and the deflection at ultimate load failure increased further. This indicated that the stiffness and ductility of the test beams gradually raised with increasing caring age. The deformation of the test beams at the same age of applied static load damage indicated that the change in spanwise deflection at static load yielding elevated slightly with increasing natural curing time for the same initial loading age. Moreover, the spanwise deflection at damage of the specimens raised significantly with increasing initial loading age [38]. All of the above studies demonstrated that the curing age affected the flexural performance of ceramsite concrete beams after creep. However, there was less research involving the effect of the curing age on the creep of ceramsite concrete beams, so it needed to be studied urgently.

*4.2. Effect of Age of Maintenance on Ceramsite Concrete Beams after Creeping*

As shown in Figure 12, the ceramsite concrete beams had a sizeable initial deflection deformation at loading. With the increase of the loading age, although the initial deflection value and the creep deflection value in the same holding time reduced gradually, the decreasing trend was less and less obvious. This was in accordance with Knaack [39] who proposed that the creep deflection decreases with the increase of the maintenance age.

As shown in Figure 13, as the age of loading increases, the reduction rate tended to zero. At this point, the total deflection reduction of the ceramsite concrete beam no longer raised significantly with the age of loading. This indicated a limit to the effect of the age of curing on the creep performance of the ceramsite concrete beams. As the load-holding time increased, the midspan deflection of the ceramsite concrete beams gradually increased. In the early stages of loading, the creep deflection rose at a faster rate and then slowly plateaued. The rate of change of the creep deflection with loading time varied for different loading ages of the ceramsite concrete beams. For the 28-d loading age, the deflection values enhanced fastest and the final total deflection values were the highest. As the loading age increased, the growth rate of the creep deflection of the ceramsite concrete beam at the

same load-holding time slowed down and the total deflection tended to reduce. At 120-d loading age, the growth rate of the creep deflection and the final deflection value of the ceramsite concrete beam at the same load-holding time was also the smallest. From the above analysis, the ceramsite concrete's strength, shrinkage creep and long-term structural deflection improved with increasing loading age.

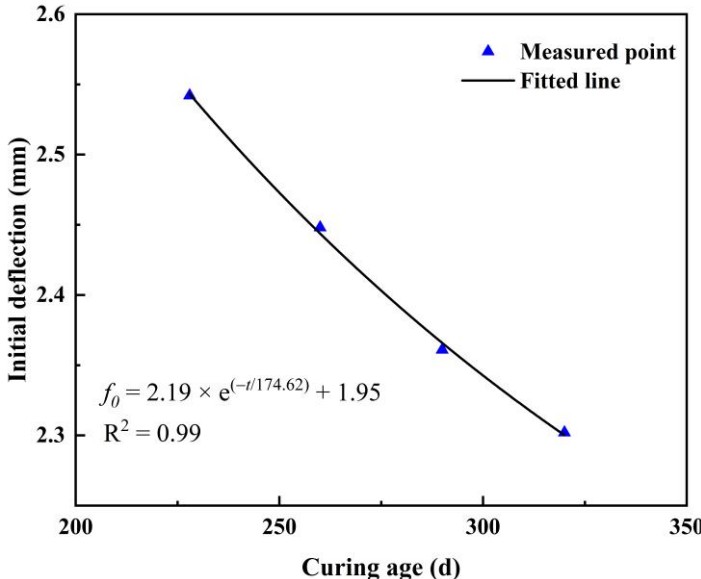

**Figure 12.** Effect of age of maintenance on the initial deflection.

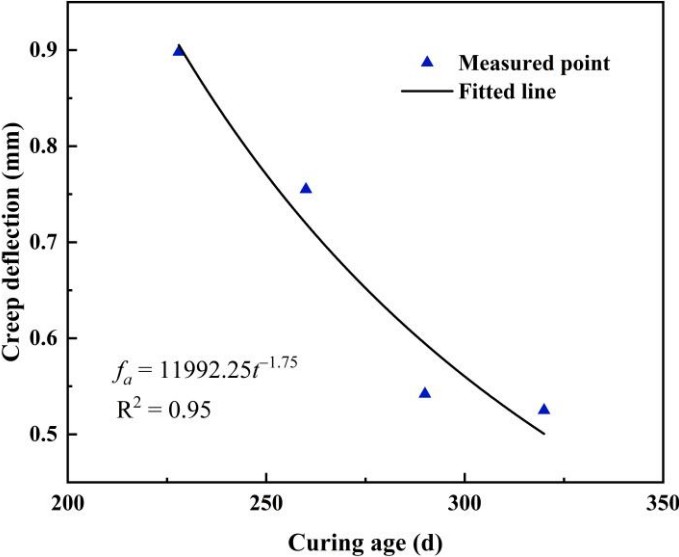

**Figure 13.** Effect of age of maintenance on creep deflection.

As displayed in Figure 14, The girders of group L2 were all loaded with 200 d of creep. The ultimate bearing capacity of the beams with a 320-d maintenance age increased by 1.81% compared with the beams with a 228 d maintenance age. Therefore, the age of curing also affected the ultimate bearing capacity of the ceramsite concrete beams subjected to creep. However, the extent of this effect was smaller at higher curing ages and the trend of increase in ultimate bearing capacity was not apparent.

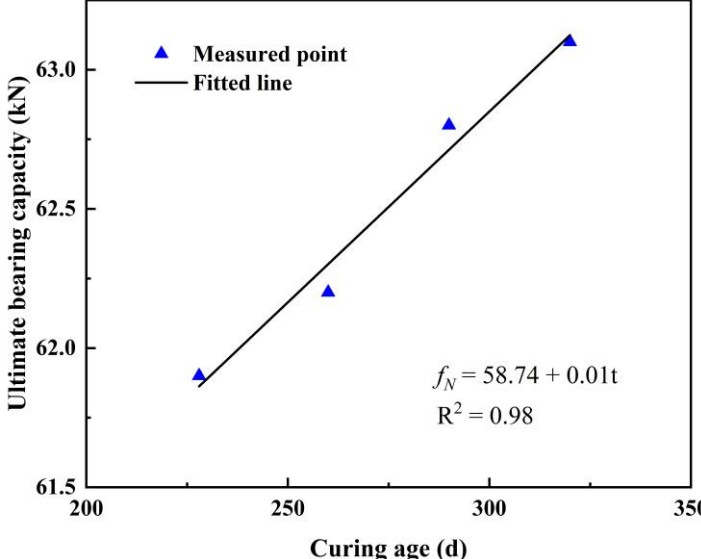

**Figure 14.** Ultimate bearing capacity curves at different maintenance ages.

Figures 15 and 16 and Table 8 suggest that as the age of curing increased, the number of cracks in the postcreep ceramsite concrete beams subjected to flexural compression damage gradually raised. In addition, it was with smaller maximum crack widths, crack depths, and average crack spacing. The ceramsite concrete beams with long curing periods had many dense cracks after bending and compression damage. Comparing with the test results of L2D1 and L2D4 beams showed the crack development pattern of the ceramsite concrete beams after creeping. This was because as the loading age of the ceramsite concrete beams increased, the internal and external moisture was continuously evaporated. In addition, the brittleness and strength class of the ceramsite concrete gradually enhanced. As a result, the number of cracks in the ceramsite concrete beams increased. The maximum crack width, depth, and average crack spacing decreased when the ultimate bearing capacity damage was reached. This indicated that as the age of maintenance increased, cracks in the beam were more likely to develop when subjected to external loads. The existing test results corroborated the crack-development pattern [40].

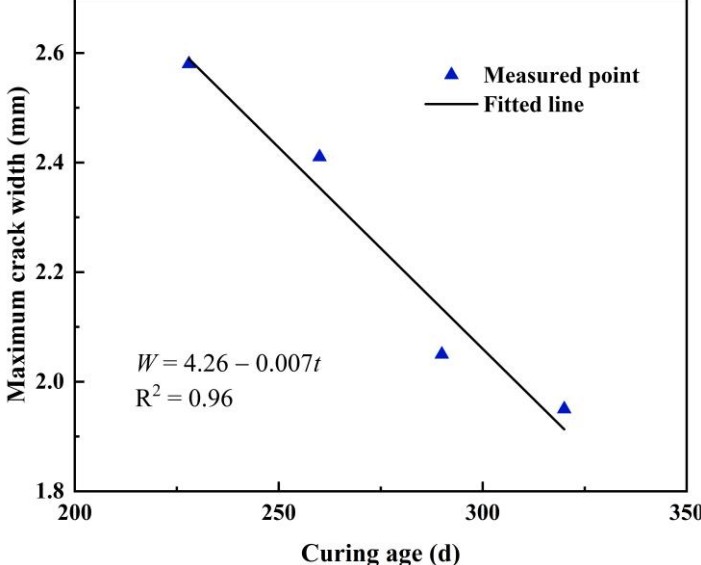

**Figure 15.** Maximum crack width curve after creep.

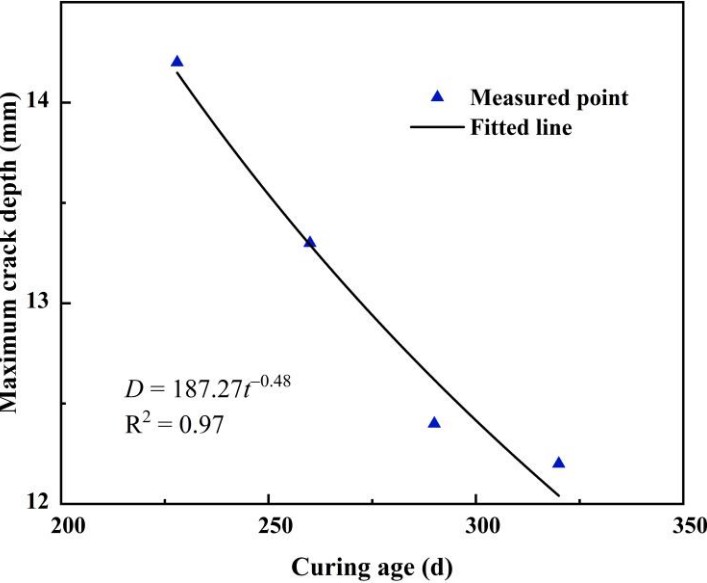

**Figure 16.** Maximum fracture depth curves after creep.

**Table 8.** Measured crack data.

| Specimen Beam Grouping Number | Extreme Load Condition | | | Maximum Crack Depth at Damage (cm) | Average Crack Spacing at Damage (cm) |
|---|---|---|---|---|---|
| | Load (kN) | Maximum Crack Width (mm) | Number of Cracks | | |
| L1D1 | 55.9 | 2.99 | 14 | 14.1 | 4.4 |
| L1D2 | 56.4 | 2.73 | 16 | 13.5 | 4.3 |
| L1D3 | 56.8 | 2.55 | 14 | 13.6 | 3.9 |
| L1D4 | 57.2 | 2.41 | 17 | 13.2 | 3.6 |
| L2D1 | 61.9 | 2.58 | 16 | 14.2 | 4.0 |
| L2D2 | 61.8 | 2.41 | 17 | 13.3 | 3.6 |
| L2D3 | 62.8 | 2.05 | 20 | 12.4 | 3.3 |
| L2D4 | 63.1 | 1.95 | 22 | 12.2 | 3.2 |

## 5. Conclusions

Based on the results and discussion presented above, the following conclusions can be obtained.

(1) Creep will increase the deformation capacity of ceramsite concrete beams and enhance the toughness and energy-absorption capacity. L2D1 exhibits higher flexural strength than L1D1, with a reinforcement value of 11%.

(2) The law of load-bearing capacity of the ceramsite concrete beams was unaffected by creep loading. However, the ultimate load-bearing capacity of the ceramsite concrete beams was promoted by creep loading. The ultimate load-bearing capacity of the test beams increased by about 10% after 200 d of creep loading at the same curing age.

(3) Creep loading affected the development of cracks. The long-term creep effect reduced the maximum crack width and crack depth. Moreover, it increased the number and average crack spacing of cracks in ceramsite concrete beams.

(4) With the increase of loading age, the initial deflection value and the creep deflection value in the same holding time also gradually decreased. In addition, the decreasing trend was less and less noticeable. As the age of loading increases, the reduction rate tended to zero.

(5) The age of curing also affected the ultimate bearing capacity of ceramsite concrete beams influenced by creep. The growth rate of the ultimate bearing capacity is faster in the early stages of curing ages, slower after 120 d, and slower and more stable in

the 228–320 d range. The ultimate load capacity of L2D4 has increased by only 1.81% compared to L2D1.

(6) The age of loading affected the ability of the ceramsite concrete beams to develop cracks after creep. The shorter the age of loading, the fewer the cracks in the ceramsite concrete beams subjected to bending damage, the greater the maximum crack width, the depth of the cracks and the average crack spacing. The longer the loading age, the more dense the cracks in the concrete beams subjected to flexural damage.

In the future, it will be planned to pour more ceramsite concrete beams. A combination of experiments and numerical simulations will be used to study the effect of curing age on the load-bearing characteristics of creep-loaded ceramsite concrete beams. The results of the study will provide a scientific reference for engineering design. In addition, sulfate erosion affects the creep of ceramsite concrete beams, which will be viewed in the subsequent research directions.

**Author Contributions:** Conceptualization, H.Z. and J.L.; methodology, J.C. and H.Z.; software, J.C.; validation, Z.F. and Y.W.; formal analysis, J.C.; investigation, Y.W.; resources, H.Z. and J.L.; data curation, Z.F., Y.W. and H.Z.; writing–original draft preparation, J.C.; writing–review & editing, Y.W. and H.Z.; visualization, P.L. and J.C.; supervision, H.Z.; project administration, H.Z.; funding acquisition, H.Z. All authors have read and agreed to the published version of the manuscript.

**Funding:** This study was funded by the National Natural Science Foundation of China (grant numbers 52178182), Hubei Provincial Excellent Young and Middle aged Science and Technology Innovation Team Project of Colleges and Universities (grant numbers T2022002).

**Institutional Review Board Statement:** Not applicable.

**Informed Consent Statement:** Not applicable.

**Data Availability Statement:** The data presented in this study are available on request from the corresponding author.

**Conflicts of Interest:** The authors declare no conflict of interest.

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
