# Peer review of "Experimental Study on Flexural Bearing Characteristics of Ceramsite Concrete Beams after Creep at Different Curing Ages"

_coatings, doi:10.3390/coatings13010158_

Round 1
Reviewer 1 Report
The citations can be improved. Novelty of the research is satisfactory and the originality is appreciated. The methods are explained and the concluding summary dealt all the results with discussions. Technically a nice work .
Author Response
请参阅附件。

Reviewer 2 Report
1-English needs revision
2-The introduction needs improvement: more information about the state of art is necessary to show the innovations of the study
3-The innovations as well as applications should be highlighted in the conclusion and abstract
4-More recent references are necessary to show the real contributions of the present study
Reviewer 3 Report
This manuscript evaluates the “Experimental Study on Flexural Bearing Characteristics of Ceramsite Concrete Beams after Creep at Different Curing Ages”. The manuscript is elaborately described and contextualized with the help of previous and present theoretical background and empirical research. All the references cited are relevant to this area of research and also adequate. The methods/analytical study are clearly stated. The result and discussion section are clearly explained. The manuscript needs Minor revision and require few modifications before the acceptance.
1. State the need of doing this research in the abstract.
2. State your recommendations after carrying out this research to the research community at the end of abstract,
3. Key words can be arranged in alphabetical order
4. Abstract need to be strengthen by citing more works.
5. Metion your research gap at the end of the introduction.
6. Table 1,2,3,5 kg-m-3. Change the unit format to kg/m3
7. Fig.3. One support is simply supported. Other support should be roller support. How it was done in your experiment?
8. Tell your research recommendation for the future research.
Reviewer 4 Report
The paper presents a good topic related to Experimental Study on Flexural Bearing Characteristics of Ceramsite Concrete Beams after Creep at Different Curing Ages. The paper should be improved according to the comments in the attached file

Author Response
Dear Reviewer:
Thank you for your valuable suggestions. We have carefully read through the comments and made proper revisions. Our responses to your questions are listed below.
1.the abstract should be rewritten to explain the methodology and the most important results.
Response: Thank you for your suggestion, we have rewritten it and shown it in blue.
2.support your results with previous studies
Response: Thank you for your suggestion, we have added the previous studies to support our results and shown it in green.
3.improve the resolution of the figure
Response: We have made the revision as you suggested and marked it in green.
4.some of points are general, make it related to your results
Response: Thank you for your suggestion, we have modified it and highlighted it in green.
Sincerely yours,
Prof. Zhu

Round 2
Reviewer 4 Report
The authors responded to the comments. I recommend to accept the paper for publication